# Exploring the gender gap in young adult mental health during COVID-19: Evidence from the UK

**Mhairi Webster**[1]*, **Sarkis Manoukian**[1], **John H. McKendrick**[2], **Olga Biosca**[1]

1 Yunus Centre for Social Business and Health, Glasgow Caledonian University, Glasgow, United Kingdom,
2 Scottish Poverty and Inequality Research Unit, Glasgow Caledonian University, Glasgow, United Kingdom

* Mhairi.Webster@gcu.ac.uk

## Abstract

### Aims

To explore the prevalence of a mental health gender gap within a young adult sample during the COVID-19 pandemic, and to identify the impact of loneliness and domestic time use on young people's, and particularly young women's mental health.

### Method

Using data from the UK Longitudinal Household Survey (UKHLS), this research examines mental health prior to the pandemic (2019) and during the pandemic (April 2020 until September 2021). A random-effects regression analysis was conducted to examine the effects of loneliness, and domestic factors across age and gender to ascertain their contribution to the mental health gender gap in a young adult population.

### Results

Average mental health decline was consistently higher for women compared to men, and young people (ages 16–24) saw a reduction in mental health twice as much as those in the oldest age category (over 65). Loneliness accounted for a share of the mental health gender gap, and a more decrease in mental health was recorded for young women experiencing loneliness, compared to older age groups. Domestic and familial factors did not have a significant impact on young people's mental health.

### Conclusions

Although across all ages and genders, mental health had returned to near pre-pandemic levels by September 2021, young people and especially women continue to have worse mental health compared to other age groups, which is consistent with pre-COVID age and gender inequalities. Loneliness is a key driver in gendered mental health inequalities during the pandemic in a young adult population.

**Data Availability Statement:** The research data are distributed by the UK Data Service. The mainstage UKHLS survey is available at: https://beta.ukdataservice.ac.uk/datacatalogue/studies/study?

id=6614, and the COVID-19 dataset is available at: https://beta.ukdataservice.ac.uk/datacatalogue/studies/study?id=8644.

**Funding:** The funding for the UKHLS/ Understanding Society COVID-19 study comes from the Economic and Social Research Council and the Health Foundation (Grant Number: ES/ M008592/1). Ipsos MORI and Kantar are responsible for conducting the fieldwork for the survey. The UKHLS, supported by the Economic and Social Research Council and multiple Government Departments, is scientifically led by the Institute for Social and Economic Research at the University of Essex. The research data are distributed by the UK Data Service. It should be emphasized that these organizations are not responsible for the analysis or interpretation of the data. This piece of research was funded and supported by the Economic Social Research Council (ESRC) and the Scottish Graduate School for Social Sciences (SGSSS) awarded to SM and MW (grant number ES/P000681/1). The funders had no role in study design, data collection and analysis, decision to publish, or preparation of the manuscript. The authors declare that they have no relevant or material financial interests that relate to the research described in this paper.

**Competing interests:** The authors have declared that no competing interests exist.

## Introduction

The COVID-19 pandemic brought about unprecedented disruptions, impacting upon, but extending far beyond, public health. Numerous studies [1–3] suggest that mental health deteriorated during the pandemic, although the extent to which adverse impact was encountered varied markedly across socioeconomic and demographic groups. Specifically, research highlighted that young people and women faced more severe mental health challenges during the pandemic [2, 4, 5]. The gender gap in mental health outcomes pre-dated the pandemic, with women being nearly twice as likely to experience a mental health problem, such as anxiety or depression, compared to men [6, 7]. This gap was exacerbated at the onset of the COVID-19 pandemic, with women experiencing a decline in mental wellbeing twice that of men in April 2020, and young women suffering worse outcomes [4].

Despite young people and women suffering worse mental health during the pandemic, it's unclear if a gender gap persists in young adults. Etheridge and Spantig [4] suggested that social isolation and caring responsibilities were responsible for most of the gender gap in mental health outcomes at the onset of the COVID-19 pandemic. However, it remains unclear whether these factors also contribute to the decline in mental health in young women, and if a gender gap in mental health remains prevalent within a young adult population.

During COVID-19, societal shifts included widespread adoption of remote work, increased reliance on digital technologies for communication and services, disruptions in education and employment, and altered social norms due to restrictions on social gatherings. Closures of workplaces and enforced social isolation disrupted social networks and increased loneliness, which further exacerbated mental health problems [8, 9]. Moreover, the closure of schools and childcare centres shifted responsibilities onto parents and caregivers, especially impacting women due to existing gender inequalities in caregiving and work [10, 11].

This paper aims to investigate the gender gap in mental health among young adults during the COVID-19 pandemic, exploring the influence of social, domestic, and caregiving roles. The research builds upon existing studies to better understand the complexities of mental health inequalities exacerbated by the pandemic's societal shifts and stresses.

## Methods

### Data

Our data originates from the UK Household Longitudinal Study (UKHLS [12–15]), also known as the 'Understanding Society' study, which is an ongoing panel that began in 2009, with around 40,000 nationally representative participants. We use both waves 10 and 11 of the mainstage Understanding Society Survey and all 9 waves from the COVID-19 survey, which was a module that ran from April 2020 to September 2021. Participants were invited to complete the COVID-19 survey if they were over the age of 16 and had participated in at least one of the previous two waves of data collection. Our overall sample with responses to the mental health outcome of interest was 141,107 observations, drawn from 26,335 unique participants. Wave 1 of our data is a pre-COVID 2019 wave comprised of responses in either wave 10 or 11 of the mainstage survey. Waves 2–10 of our data comprise of each COVID wave starting in April 2020 (wave 2) and ending in September 2021 (wave 10)- see S1 Appendix for details.

### Methodology

The main outcome of interest is mental health, which was derived from the General health questionnaire (GHQ-12), which tests common mental health and wellbeing indicators such as loss of sleep, enjoyment of day-to-day activities, and ability to deal with problems. Importantly,

it asks participants to score each question relative to their "usual" wellbeing and as such includes a reference point for participants to evaluate their current feelings. Each question is answered on a scale of 0 (least distressed) to 3 (most distressed), concluding with a total score of between 0 and 36. A higher GHQ score relates to higher psychological distress; a lower score indicates lower psychological distress. For details on the specific GHQ, see S2 Appendix.

The GHQ-12 has been shown to be a valid and reliable unidimensional measure of generalised mental health, with age, gender and education not affecting the validity of the measurement [16–18]. It has been widely used in social science research [19–21] including research on mental health during COVID-19 [1]. Most importantly, this measure is included in all UKHLS mainstage and COVID-19 waves, providing a reliable and consistent measure of mental health both prior to and during the pandemic.

We extracted data on loneliness and domestic time use to understand the impact of social isolation and caregiving respectively. Loneliness was derived from the question "In the last 4 weeks, how often did you feel lonely?" with responses being "Never/Hardly", "Some of the time" and "Often". Domestic time use was derived from hours spent cleaning and hours caring for children per week. Individuals with missing data regarding the variables of interest (including refusal and "don't know" responses) were excluded from the analysis, see S3 Appendix [14, 15].

The models include demographic variables such as age (grouped), sex, ethnicity (white, or belonging to a black or other ethnic minority(BME)), highest level of education (degree, other higher education, A evel, GSCE, other qualification and no qualifications), employment status (employed, self-employed, unemployed retired, student and other), household composition (living alone, living with other adults, single parents, and households with both adults and children), marital status (single/ divorced/widowed, married and in a relationship) and if an individual had a long-term illness. Two geographical indicators were extracted, i.e.UK region (England, Scotland, Wales and Northern Ireland) and urban/rural status.

Firstly, we present descriptive statistics of mental health variation by demographic and socioeconomic characteristics and a subgroup of mean GHQ at three points in time—2019 (pre-pandemic), April 2020 (Early Pandemic) and September 2021 (End of the pandemic) and an overall pandemic mean GHQ (from April 2020—September 2021), which was treated as a single wave for this analysis. Secondly, we analyse specific changes in mean GHQ across the pandemic by age and gender. Thirdly, we examine the effects of both loneliness and domestic time use on GHQ for different age groups (see S4 Appendix). Finally, a random-effects GLS regression was carried out to analyse the impact of both loneliness and domestic time use on GHQ, and identify how this varied by sex and age. The final model (VI) includes all relevant variables and controls, including sociodemographic characteristics (see S5 Appendix for the full model). All analysis was conducted through Stata-17 [22].

## Results

Descriptive analysis of GHQ in September 2021 highlights that the mental health gender gap persists, with women experiencing a higher level of psychological distress compared to men, indicated by a higher GHQ score. Young people and women experienced consistently higher psychological distress throughout the pandemic, with the greatest increases being registered during periods of lockdown in April 2020 (wave 2) and November 2020 (wave 7). The gender gap in mean GHQ scores was most pronounced at the outset of the pandemic in April 2020, gradually narrowing to a difference of 1.35 between men and women by September 2021. However, as of September 2021, a significant mental health age gap persisted between young adults (ages 16–24) and older adults (ages 65+), with a mean difference of 2.56 for young men

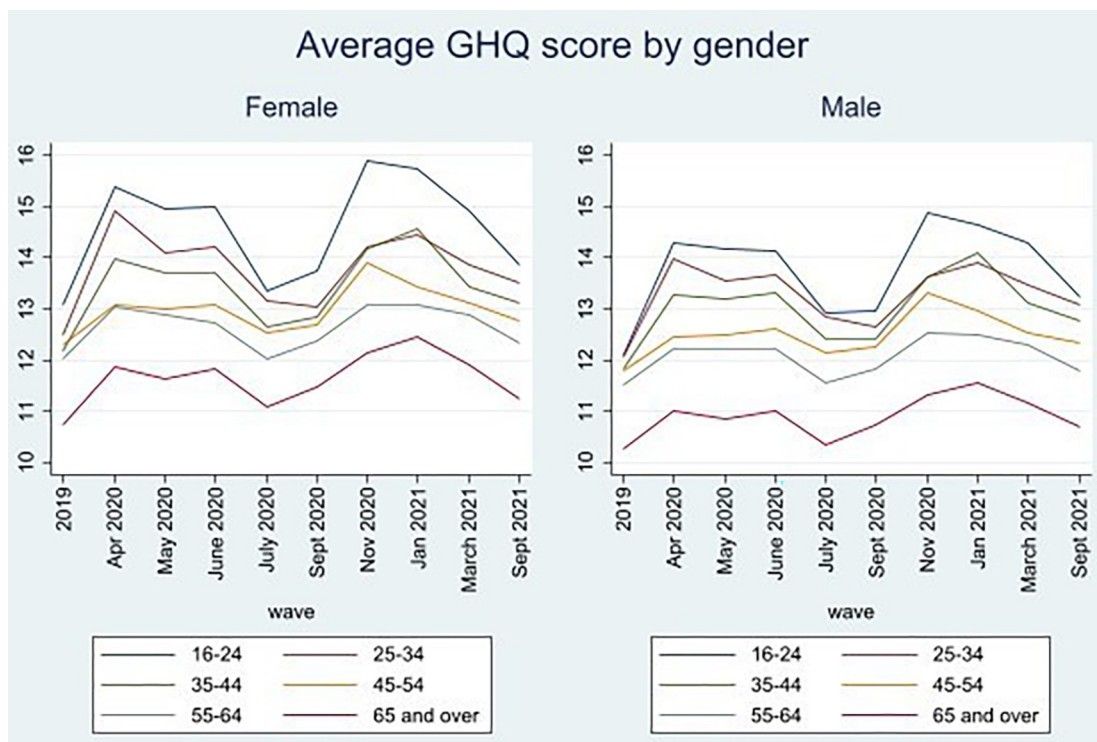

**Fig 1. Mean male and female GHQ scores across time.** Data from UKHLS: waves 10 and 11 of the mainstage UKHLS study, and waves 1–9 of the COVID-19 survey.

and 2.63 for young women compared to their older counterparts, respectively. However, mental health recovered to near pre-pandemic levels for all ages and genders by September 2021 (see Figs 1 and 2), despite a persistent age and gender gap.

Table 1 reports mean psychological distress (GHQ) by various socioeconomic and demographic characteristics, loneliness and domestic time use at baseline (2019), at the beginning of the pandemic (April 2020), the end of the pandemic (September 2021) and a difference between GHQ scores from these start and end dates. Overall, women, younger people, people who were unemployed, single parents, people living in an urban area, and people with a long-term limiting illness had higher GHQ scores than their comparator groups throughout the pandemic. Loneliness resulted in extreme increases in GHQ score, with those who reported often feeling lonely having an average GHQ scores 12.5 points higher compared to those who hardly or never felt lonely. GHQ scores for those who felt lonely also increased across the pandemic, unlike other demographics who experienced a recovery in mental health from April 2020 to September 2021. The prevalence of young women who were often feeling lonely was highest during periods of lockdown, with the peak occurring during the first lockdown (April 2020). Conversely, a higher proportion of young men experienced frequent loneliness during the third lockdown period in the UK (January 2021), nearly matching the levels reported by young women during the same period (see S6 Appendix).

Model I (Table 2) show that young adults are likely to experience an increase of 0.66 to their GHQ score when compared to older adults, without controlling for any factors other than age. Women's GHQ score was more likely to be 1.57 points higher compared to men. Young women similarly demonstrated an increased probability of having a higher GHQ score by 0.9 points compared to men and women over the age of 25. Model II shows that loneliness

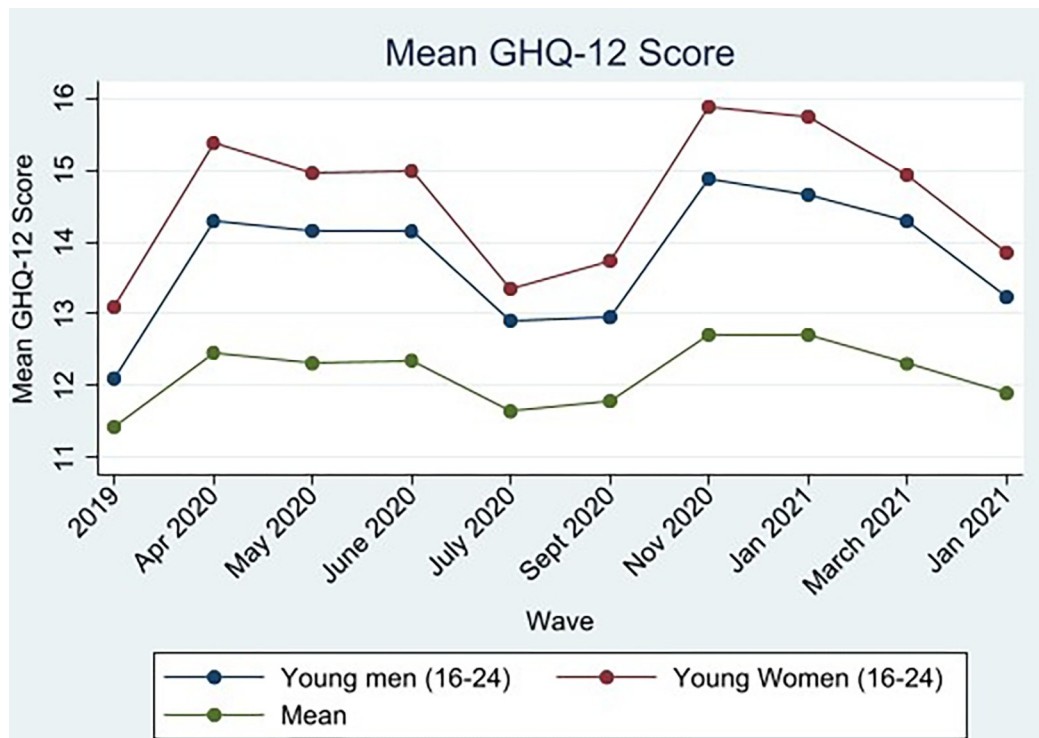

**Fig 2. Mean young male and female GHQ scores across time.** Data from UKHLS: waves 10 and 11 of the mainstage UKHLS study, and waves 1–9 of the COVID-19 survey.

had a strong relationship with GHQ, with those who felt sometimes or often lonely having a higher probability to experience increased psychological distress, indicated by a higher GHQ score. This model also lowered the effect of gender and particularly age, lowering the probability for young adults and women to experience an increased GHQ score. Loneliness also accounts for a large amount of the variation in GHQ scores (34%). Time spent on childcare does have a clear relationship with GHQ, however cleaning for between 6–10 hours per week appears to have a small positive effect on GHQ when compared to cleaning for only 0–6 hours per week. However, neither domestic time use variables shows an effect on the age and gender coefficients.

When controlling for sociodemographic factors in model VI (education, ethnicity, health, employment status, marital status, housing composition, geographical indicators and time), we see that the effect of being a young adult has decreased further to -0.65, with loneliness still having a significant impact on GHQ scores. The coefficient for being female also decreases to 0.97. Living in a household with children or having a disability was also associated with a higher probability of having a higher GHQ score compared to those living with adults only and to individuals who are living without a disability respectively.

## Discussion

This research finds that mental health steeply declined during the early months of the pandemic, before recovering when lockdown restrictions were eased (waves 4–6 = June-September 2020), which is similar to previous research [2, 23]. During the beginning of June and throughout the summer period, strict lockdowns were lifted across the UK, and schools and non-essential shops introduced a phased re-opening. National restrictions were relaxed, and

**Table 1. Mean GHQ scores (2019- September 2021).**

| Demographic | Overall sample n (%) | GHQ in 2019 | GHQ in April 2020 | GHQ in Sept 2021 | Difference April 2020-September 2021 |
|---|---|---|---|---|---|
| **Sex** | | | | | |
| Male | 59,103 (41.9) | 10.73 (5.25) *** | 11.21 (5.44) *** | 11.09 (5.45) *** | -0.12 |
| Female | 81,915 (58.1) | 11.96 (5.87) *** | 13.35 (6.31) *** | 12.44 (6.07) *** | -0.91 |
| **Age** | | | | | |
| 16–24 | 9,469 (6.7) | 12.1 (6.23) *** | 14.29 (6.55) *** | 13.23 (6.88) *** | -1.06 |
| 25–34 | 13,664 (9.7) | 12.01 (5.98) *** | 13.99 (6.77) *** | 13.07 (6.46) *** | -0.92 |
| 35–44 | 20,259 (14.4) | 11.83 (5.77) *** | 13.26 (6.36) *** | 12.78 (6.37) *** | 0.52 |
| 45–54 | 27,510 (19.5) | 11.81 (5.85) *** | 12.46 (5.99) *** | 12.33 (5.98) *** | -0.13 |
| 55–64 | 30,612 (21.7) | 11.48 (5.79) *** | 12.21 (6) *** | 11.79 (5.95) ****** | -0.42 |
| 65+ | 39,590 (28) | 10.25 (4.77) *** | 11.01 (5.08) *** | 10.67 (4.82) *** | -0.34 |
| **Ethnicity** | | | | | |
| White | 117,942 (88.9) | 11.39 (5.44) * | 12.34 (5.95) *** | 11.81 (5.82) *** | -0.53 |
| Black or ethnic minority (BME) | 14, 696 (11.1) | 11.58 (6.1) * | 13.43 (6.79) *** | 12.53 (6.29) *** | -0.9 |
| **Higher Education** | | | | | |
| No qualification | 6,250 (4.8) | 11.48 (5.69) * | 12.41 (6.3) ** | 12.02 (5.89) | -0.39 |
| Other Qualification | 9,162 (7) | 11.4 (5.7) * | 12.03 (6.04) ** | 11.84 (5.98) | -0.19 |
| GSCE | 23,672 (18) | 11.66 (5.86) * | 12.5 (6.15) ** | 12.22 (6.16) | -0.28 |
| A-levels | 26,886 (20.4) | 11.54 (5.78) * | 12.58 (6.13) ** | 11.97 (6.05) | -0.61 |
| Other Higher Degree | 18,158 (13.8) | 11.36 (5.66) * | 12.49 (6.06) ** | 11.9(5.74) | -0.59 |
| Degree | 47,461 (36) | 11.19 (5.33) * | 12.45 (5.94) ** | 11.69 (5.66) | -0.76 |
| **Relationship Status** | | | | | |
| Single/ Divorced/ Widowed | 37,024 (28) | 12.1 (6.25) *** | 13.57 (6.6) *** | 12.84 (6.67) *** | -0.73 |
| Married | 94,573 (71.4) | 11.04 (5.25) *** | 12 (5.76) *** | 11.53 (5.51) *** | -0.47 |
| In a relationship | 884 (0.6) | 11.67 (5.38) *** | 12.95 (6.45) *** | 11.49 (4.85) *** | -1.46 |
| **Employment Status** | | | | | |
| Employed | 47,432 (49.2) | 11.28 (5.31) *** | 12.51 (5.99) *** | 12.03 (5.83) *** | -0.48 |
| Self-Employed | 7,693 (8%) | 10.61 (4.89) *** | 12.38 (6.16) *** | 11.46 (5.44) *** | -0.92 |
| Unemployed | 2,310 (2.4) | 14.68 (7.47) *** | 13.82 (6.77) *** | 13.81 (6.99) *** | -0.01 |
| Retired | 27,937 (29) | 10.34 (4.71) *** | 11.23 (5.19) *** | 10.74 (4.92) *** | -0.49 |
| Student | 4,022 (4.2) | 11.88 (6.03) *** | 14.58 (6.77) *** | 13.15 (6.92) *** | -1.43 |
| Other | 6,965 (7.2) | 14.46 (7.49) *** | 14.94 (7.28) *** | 14.49 (7.49) *** | -0.45 |
| **Household Composition** | | | | | |
| Living alone | 19,776 (14.9) | 11.66 (5.98) *** | 13 (6.34) *** | 12.39 (6.36) *** | - 0.61 |
| Living with other adults | 71,762 (54.1) | 11.1 (5.38) *** | 11.96 (5.79) *** | 11.4 (5.57) *** | -0.56 |

*(Continued)*

**Table 1.** (Continued)

| Demographic | Overall sample n (%) | GHQ in 2019 | GHQ in April 2020 | GHQ in Sept 2021 | Difference April 2020-September 2021 |
|---|---|---|---|---|---|
| Single Parents | 2,952 (2.2) | 13.64 (6.91) *** | 14.28 (6.99) *** | 13.93 (7.15) *** | -0.35 |
| Adults and children | 38,197 (28.8) | 11.59 (5.68) *** | 12.94 (6.22) *** | 12.38 (5.98) *** | -0.56 |
| **Region** | | | | | |
| England | 107, 319 (80.9) | 11.44 (5.63) *** | 12.5 (6.08) | 11.8 (5.81) ** | -0.7 |
| Scotland | 11,656 (8.8) | 11.12 (5.77) *** | 12.54 (6.04) | 12.05(6.17) ** | -0.49 |
| Wales | 7,867 (5.9) | 11.72 (5.69) *** | 12.62 (6.19) | 12.53 (6.19) ** | -0.09 |
| Northern Ireland | 5,814 (4.4) | 11.14 (5.43) *** | 12.31 (5.63) | 12.37 (6.04) ** | 0.06 |
| **Location** | | | | | |
| Urban | 97,208 (74) | 11.55 (5.74) *** | 12.63 (6.16) *** | 12 (5.95) *** | -0.63 |
| Rural | 34,064 (26) | 11.04 (5.31) *** | 12.01 (5.6) *** | 11.56 (5.66) *** | -0.45 |
| **Disability** | | | | | |
| No long-standing illness or disability | 83,689 (63.2) | 10.44 (4.82) *** | 11.97 (5.68) *** | 11.25 (5.4) *** | -0.72 |
| Having a long-standing illness or disability | 48,713 (36.8) | 13.08 (6.48) *** | 13.34 (6.6) *** | 13 (6.49) *** | -0.34 |
| **Loneliness** | | | | | |
| Hardly or Never | 88,312 (63.4) | 9.43 (4) *** | 10.21 (4.44) *** | 9.81 (3.95) *** | -0.4 |
| Some of the time | 42,016 (30.2) | 13.45 (5.56) *** | 15.12 (5.64) *** | 14.88 (5.69) *** | -0.24 |
| Often | 8,868 (6.4) | 19.02 (7.42) *** | 21.41 (7.17) *** | 23.35 (7.83) *** | 1.94 |
| **Hours spent on childcare** | | | | | |
| 0 hours | 49,090 (34.8) | 11.37 (5.63) | 12.25 (5.97) ** | 11.73 (5.77) ** | -0.52 |
| 1–15 hours | 50,970 (36.1) | 11.43 (5.62) | 12.47 (6) ** | 11.87 (5.88) ** | -0.6 |
| >15 hours | 41,046 (29.1) | 11.46 (5.68) | 12.68 (6.22) ** | 12.1 (5.98) ** | -0.58 |
| **Hours spent on cleaning** | | | | | |
| Less than 6 | 41,216 (29.2) | 11.43 (5.63) | 12.11 (6.06) *** | 11.91 (6.07) *** | -0.2 |
| 6–10 hours | 43,275 (30.7) | 11.38 (5.66) | 12.09 (5.76) *** | 11.38 (5.48) *** | -0.71 |
| >10 hours | 56,615 (40.1) | 11.38 (5.56) | 12.88 (6.18) *** | 12.06 (5.82) *** | -0.82 |

Notes

1. Unweighted N, total sample = 141,107

2. Wilcoxon Rank-Sum Test used to test for significance of difference between groups of 2 (sex, ethnicity, long-standing illness). Kruskal-Wallis used to test for differences between groups of 3 or more: * p<0.05, ** p<0.01, *** p<0.001.

3. GHQ, 12 item General Health Questionnaire

local authorities were given additional powers to enforce social distancing or regional lock-downs in their local constituency. By August, the "Eat out to help out" scheme was introduced to support and stimulate the service industry, and lockdown restrictions were further loosened, with bowling alleys and indoor theatres and reopening. However, by mid-September, many of these restrictions returned, and a second lockdown was announced in England on the 31st of

**Table 2. Random-effects regression analysis (Models I-IV) showing the change in GHQ score associated with age and gender.**

|  | I | II | III | IV |
|---|---|---|---|---|
|  | β (sd.error) | β (sd.error) | β (sd.error) | β (sd.error) |
| **Young Adult** | 0.66 (0.08) *** | -0.4 (0.18)* | 0.7 (0.23)** | -0.65 (0.03)** |
| **Female** | 1.57 (0.78)* | 0.99(0.06)*** | 1.59 (0.08)*** | 0.97 (0.06)*** |
| **Young Woman** | 0.9 (0.27)** | 0.85 (0.21)*** | 0.95 (0.28)** | 0.89 (0.22)*** |
| **Controls:** |  |  |  |  |
| **Loneliness** |  |  |  |  |
| Sometimes |  | 3.26 (0.03) *** |  | 3.2 (0.03) *** |
| Often |  | 8.51 (0.06) *** |  | 8.34 (0.06) *** |
| **Childcare** |  |  |  |  |
| 1–15 hours |  |  | 0.02(0.03) | 0.02(0.03) |
| > 15 hours |  |  | 0.03 (0.03) | 0.03 (0.03) |
| **Clean** |  |  |  |  |
| 6–10 hours |  |  | -0.15 (0.04) *** | -0.14 (0.04)*** |
| >10 hours |  |  | -0.04 (0.04) | -0.04(0.04) |
| **Education** |  |  |  |  |
| Other Higher degree |  |  |  | -0.18 (0.09) * |
| A-Level |  |  |  | -0.19(0.08)* |
| GCSE |  |  |  | -0.19 (0.08)* |
| Other Qual |  |  |  | -0.55 (0.12)*** |
| No Qual |  |  |  | -0.55 (0.14)*** |
| **Household Composition** |  |  |  |  |
| Adults only |  |  |  | 0.5 (0.11) *** |
| Single parent |  |  |  | 0.84 (0.2)*** |
| Adults and children |  |  |  | 1.17 (0.12)*** |
| **Urban** |  |  |  | 0.26 (0.07)*** |
| **Disabled** |  |  |  | 1.18 (0.06) *** |
| **Constant** | 11.4 (0.06) *** | 10.24 (0.05) *** | 11.41 (0.07) *** | 9.86(0.12) *** |
| Observations | 117,866 | 117,833 | 117,845 | 111,567 |
| $R^2$ | 0.02 | 0.34 | 0.02 | 0.34 |

Notes

1. Data from the UKHLS COVID module. Random effect model building for GHQ Likert, age, gender and relevant explanatory variables.

2. Reference categories are never/ hardly lonely, 0 hours caring for children, 0–6 hours of cleaning, living alone, being in employment, having a degree, being single/divorced/widowed, living in England.

3. Binary measures for age, gender and an age-gender interaction, ethnicity, living in an urban area and having a disability.

4. A robustness check was conducted with a standardized GHQ as the dependent variable- all other variables remained the same.

5. * = $p<0.05$; ** = $p<0.01$; *** = $p<0.001$.

6. Model I is regressed against only binary variables for age, gender and an interaction. Model II and III builds on this to regress against loneliness and domestic time use respectively. Model IV includes all explanatory variables while controlling for ethnicity, education, marital status, disability, household composition, employment status, geographical indicators (UK region, and an urban/rural indicator), and time.

October, with devolved nations shortly adopting localized lockdowns in areas with a high coronavirus infection rate. Given the context of the changing social and economic landscape during the pandemic, it is unsurprising that as lockdown restrictions eased, people's mental health generally increased. At this time, many would have been able to visit their families and friends for the first time in months or be able to return to work after months of uncertainty. This clearly highlights the impact of these restrictive measures on mental health.

Consistent with broader patterns observed in other wellbeing and mental health metrics, GHQ scores typically have lower levels during the summer months (April to October) and higher during the winter period, as documented by Banks & Xu [2]. However, previous studies using the same panel sample conclude that the rise in GHQ scores was unlikely to be due to seasonal differences or year to year variation [24]. Additionally, by treating time as a nested level within individuals, this method allows for the examination of within-person changes while also capturing any potential effects of seasonal trends. Because of this approach we can be confident that any changes in GHQ are not seasonal and must be the result of other factors.

Long term trends show a second steep increase in mental health issues during the second and third periods of lockdown in the UK (waves 7–9 = November 2020 to March 2021), before recovering again in September 2021, demonstrating a similar pattern to the mental health decline at the onset of the first lockdown. Overall, mental health has returned to near pre-pandemic levels by 2021, though young people and women continue to report higher levels of psychological distress compared to men and older adults, suggesting that the pandemic did not alter relative risk across age and gender. These results are in line with previous research [3] which also found that young women were more vulnerable to mental health decline during periods of lockdown. The mental health gender gap persists until September 2021 but has reduced from a 1.1-point average difference between young men and young women's GHQ score, to a 0.6-point difference, suggesting that the gender gap was much larger at the beginning of the pandemic and has narrowed over time. This gap is smaller than it was in 2019, however this may be due to young men's mental health being slower to recover and seeing a greater decline than women compared to their pre-pandemic values. This may indicate a broader issue wherein men may encounter difficulties in recovering their mental well-being following the COVID-19 pandemic, especially considering the existing stigma and gender norms surrounding their access to mental health assistance [25]. Despite this, the mental health gender gap continued to be a present issue during the pandemic, highlighting ongoing inequalities in young people's mental health outcomes.

We find that during the pandemic, young women are more likely to experience a reduction in mental health when lonely. As noted by previous research [4], this effect may be due to women's larger support networks (for example having a greater number of close friends), which may greatly impact their mental health when these support networks are disrupted by restrictions on social gatherings [26, 27]. Similar results were found by previous research that women were more vulnerable to mental health decline during periods of lockdown [3, 28]. Though this research did not investigate social support through the role of friendships, it did identify that young people's and young women's mental health was more impacted during periods of lockdown, with a greater reduction of mental health at the beginning of periods of lockdown, and a greater increase after lockdown restrictions were lifted. Additionally, loneliness had a more severe effect on young people's psychological distress, as evidenced by an increased probability of a higher GHQ score for individuals who reported that they "often" felt lonely.

While extensive research has explored the gendered effects of domestic labor and gender inequality on women's mental health amidst the pandemic [4, 29], our study indicates that domestic inequality is unlikely to be the primary cause of the mental health gender gap among young adults. An increased amount of time spent on cleaning or childcare did not affect young people's GHQ scores in comparison to older age groups, which can be explained by the small proportion of young adults with childcare responsibilities- only 7.7% of those ages 16–24 reported caring for children for more than 15 hours per week. In comparison, the proportion of the next age bracket, 25–34 years old's, was 48.6% (See S7 Appendix). Overall, it is unlikely that childcare responsibilities and domestic duties are responsible for the mental health gender gap within a young adult population.

## Implications

Our study contributes to the body of research examining loneliness and mental health. However, it specifically notes the impact on young people, particularly on young women. This is important, as this is a group with the poorest mental health outcomes during the pandemic, and as such may require specific responses and interventions. It also may emphasise the importance of social support during times of crisis, particularly for vulnerable young people. During the pandemic many of these social networks were severely restricted (i.e., friendships, education, family) due to lockdown measures. Policymakers may need to consider the mental health implications of social restrictions and lockdowns when implementing public health measures, and moving forward, it's imperative to prioritize mental health support and resources alongside physical health initiatives. This could involve integrating mental health services into crisis response plans, and fostering community support networks to mitigate the psychological impact of loneliness during times of crisis. Balancing the need for physical health measures with the potential social and mental health consequences is crucial for developing holistic public health strategies. Furthermore, this underscores the significance of social policy in shaping mental health outcomes, potentially influencing future policy development and decision-making by policymakers.

## Limitations

Although the UKHLS COVID-19 survey provides rich data throughout key moments of the pandemic, there are known limitations of the dataset. The non-response rate across the COVID-19 survey waves was around 70%, which may affect the validity of the presented result. However, the demographic characteristics of our sample are similar in proportion to the overall sample, including non-respondents to the COVID-19 survey.

## Conclusions

Social factors, specifically loneliness, contribute to the mental health gender gap within the young adult population. Young people, especially young women, experienced the greatest reductions in their mental health during strict lockdown periods, likely due to loneliness and social isolation. It is imperative when addressing future pandemic responses to strike a balance between protecting public health and maintaining the mental health of vulnerable groups. Promotion of policies to address loneliness within a younger population, and especially in young women, may help young people's mental health to recover in the aftermath of the pandemic. Additionally, long-term trends in mental health should be monitored. Although this research suggests that mental health within the UK population has recovered to pre-pandemic levels, it is unknown what the longer-term effects of the pandemic will be on mental health and other health behaviours. Additionally, a gender gap persists in mental health outcomes beyond the beginning of the pandemic. This gap must be addressed through promotion of gender equality and equal access to mental health services. Young men's mental health also seems to have been slower to rebound, with a more pronounced decline and slower recovery compared to young women, relative to their pre-pandemic levels. This suggests that while progress has been made in narrowing the gender gap, challenges persist, and further efforts may be needed to fully address and understand the complexities of mental health disparities during and after the pandemic.

## Supporting information

**S1 Appendix. Appendix A- Source data.**
(DOCX)

**S2 Appendix. GHQ-12 survey question design.**
(DOCX)

**S3 Appendix. Missing data.**
(DOCX)

**S4 Appendix. Random-effects regression subgroup analysis of the impact of loneliness and domestic time use on GHQ.**
(DOCX)

**S5 Appendix. Framework for regression model.**
(DOCX)

**S6 Appendix. Proportion of individuals feeling "often lonely" by age and gender.**
(DOCX)

**S7 Appendix. Proportion of loneliness, hours spent on childcare and hours spent on cleaning by age and gender.**
(DOCX)

**S1 Dataset. Merge dataset.**
(DO)

## Acknowledgments

The UKHLS, supported by the Economic and Social Research Council and multiple Government Departments, is scientifically led by the Institute for Social and Economic Research at the University of Essex. The research data are distributed by the UK Data Service. It should be emphasized that these organizations are not responsible for the analysis or interpretation of the data.

## Author Contributions

**Conceptualization:** Mhairi Webster, Sarkis Manoukian.

**Data curation:** Mhairi Webster.

**Formal analysis:** Mhairi Webster.

**Funding acquisition:** Sarkis Manoukian.

**Methodology:** Mhairi Webster, Sarkis Manoukian.

**Project administration:** Sarkis Manoukian.

**Supervision:** Sarkis Manoukian, John H. McKendrick, Olga Biosca.

**Visualization:** Mhairi Webster.

**Writing – original draft:** Mhairi Webster.

**Writing – review & editing:** Sarkis Manoukian, John H. McKendrick, Olga Biosca.

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
