## [Decision Letter · Decision Letter 0]

13 Sep 2024

PONE-D-24-22134Exploring the Gender Gap in Young Adult Mental Health during COVID-19: Evidence from the UKPLOS ONE

Dear Dr. Webster,

Thank you for submitting your manuscript to PLOS ONE. After careful consideration, we feel that it has merit but does not fully meet PLOS ONE’s publication criteria as it currently stands. Therefore, we invite you to submit a revised version of the manuscript that addresses the points raised during the review process.

We look forward to receiving your revised manuscript.

Kind regards,

Kamlesh Kumar Sahu

Academic Editor

PLOS ONE

“The funding for the UKHLS/Understanding Society COVID-19 study comes from the Economic and Social Research Council and the Health Foundation. Ipsos MORI and Kantar are responsible for conducting the fieldwork for the survey. The UKHLS, supported by the Economic and Social Research Council and multiple Government Departments, is scientifically led by the Institute for Social and Economic Research at the University of Essex.

The research data are distributed by the UK Data Service. It should be emphasized that these organizations are not responsible for the analysis or interpretation of the data.

This research was funded and supported by the Economic Social Research Council (ESRC) and the Scottish Graduate School for Social Sciences (SGSSS).”

The authors declare that (s)he has no relevant or material financial interests that relate to the research described in this paper.

“The funding for the UKHLS/Understanding Society COVID-19 study comes from the Economic and Social Research Council and the Health Foundation. Ipsos MORI and Kantar are responsible for conducting the fieldwork for the survey. The UKHLS, supported by the Economic and Social Research Council and multiple Government Departments, is scientifically led by the Institute for Social and Economic Research at the University of Essex. The research data are distributed by the UK Data Service. It should be emphasized that these organizations are not responsible for the analysis or interpretation of the data.”

“The funding for the UKHLS/Understanding Society COVID-19 study comes from the Economic and Social Research Council and the Health Foundation. Ipsos MORI and Kantar are responsible for conducting the fieldwork for the survey. The UKHLS, supported by the Economic and Social Research Council and multiple Government Departments, is scientifically led by the Institute for Social and Economic Research at the University of Essex.

The research data are distributed by the UK Data Service. It should be emphasized that these organizations are not responsible for the analysis or interpretation of the data.

This research was funded and supported by the Economic Social Research Council (ESRC) and the Scottish Graduate School for Social Sciences (SGSSS).”

Additional Editor Comments:

Reviewers' comments:

Reviewer's Responses to Questions

**Comments to the Author**

1. Is the manuscript technically sound, and do the data support the conclusions?

Reviewer #1: Yes

Reviewer #2: Yes

2. Has the statistical analysis been performed appropriately and rigorously? 

Reviewer #1: Yes

Reviewer #2: Yes

3. Have the authors made all data underlying the findings in their manuscript fully available?

Reviewer #1: Yes

Reviewer #2: No

4. Is the manuscript presented in an intelligible fashion and written in standard English?

Reviewer #1: Yes

Reviewer #2: Yes

5. Review Comments to the Author

Reviewer #1: Abstract:

It is good if the result has statistical result

Introduction:

It can be good to also describe the gender gap in mental health in other countries and regions. There could be factors, such as culture, social habit, or race that influence the gender gap in mental health during Covid-19.

Methods:

1. It would interest the readers to read more description about the survey.

2. It is also still important to mention important aspects of the methods such as, sampling techniques used, did all region sampled in the national study.

3. "141,107 observations from 26,335 unique participants" Does the sample size bigger than national study data? It is also different from the total number of sample in table 1 with 141,104 samples, 3 missing data.

Reviewer #2: Using a subsample of the UK Longitudinal Household Survey, the authors track changes in mental heath among young adults during Covid pandemic. Employing random-effects regression analyses, they examine the impact of loneliness and domestic factors across age and gender to ascertain their contribution to the gender gap in mental health in this population.

The paper reports deterioration in mental health among women relative to men, and among young adults relative to those 65+. The authors also found that loneliness played a role in widening the mental health gender gap, noting that, compared to older age groups, the mental health of young women was influenced by loneliness. The authors conclude that young adults, especially women, continue to have worse mental health compared to other age groups, with loneliness being a key driver in gendered mental health disparities.

The research questions explored in this study – if repeated in a number of robust studies – can be of significance for the implementation of public health measures in times of pandemics and/or other (ecological) crises.

With that in mind, this reviewer has the following to remark:

1. The Abstract:

It would be of interest if the authors include the statistics to give the reader a clearer idea of their findings from the outset.

2. Results section:

The authors have reported β and p of their random-effects regression

models. It would have been informative to see the 95% CIs.

Another point has to do with the models and the distribution of random effects. While the normality assumption is sometimes hard to ascertain, it is still important to check the extent to which the estimates could be biased if the assumption is not met. In this context, did the authors assess this issue? Did they consider potentially using models with non-normal random effects?

3. Limitation section:

A last point involves the limitations. The only limitation mentioned is that the non-response rate across the survey waves was around 70%. This paper would be stronger if it incorporates other obvious limitations.

I hope this review is helpful and wish the authors the very best with their research!

6. PLOS authors have the option to publish the peer review history of their article (what does this mean?). If published, this will include your full peer review and any attached files.

Reviewer #1: No

Reviewer #2: No

---

## [Author Response · Author response to Decision Letter 0]

25 Oct 2024

Response to reviewers R1

PONE-D-24-22134

Exploring the Gender Gap in Young Adult Mental Health during COVID-19: Evidence from the UK

Dear Dr Kamlesh Kumar Sahu and reviewers, 

On behalf of my colleagues, I would like to extend my thanks for your time and effort in reviewing our manuscript. I am pleased to submit a revised manuscript of the original research article, “Exploring the Gender Gap in Young Adult Mental Health during COVID-19: Evidence from the UK”. 

We found the reviewers' comments and suggestions highly helpful and consistent, and we have made every effort to address them thoroughly. We believe the revisions have sharpened the manuscript’s focus and significantly improved its quality and readability. 

As required, in addition to this letter we have included the following items in the resubmission: 

• A marked-up copy of our manuscript that highlights changes in red font made to the original version, labelled 'Revised Manuscript with Track Changes'. 

• An unmarked version of your revised paper without tracked changes. 

Additionally, we submit two supporting information files alongside this re-submission. 

• S1- Appendix A-Source Data. This was uploaded incorrectly during the initial submission. The previous incorrect S1-Appendix A has been removed. 

• S8- Supporting file. This is a do-file with code to merge the UKHLS and COVID-19 datasets available from the UK Data Service. 

In addition to the changes made to the original manuscript, we have edited the following sections of the online submission: 

1. Financial Disclosure

 “The funding for the UKHLS/Understanding Society COVID-19 study comes from the Economic and Social Research Council and the Health Foundation. Ipsos MORI and Kantar are responsible for conducting the fieldwork for the survey. The UKHLS, supported by the Economic and Social Research Council and multiple Government Departments, is scientifically led by the Institute for Social and Economic Research at the University of Essex.

The research data are distributed by the UK Data Service. It should be emphasized that these organizations are not responsible for the analysis or interpretation of the data.

This research was funded and supported by the Economic Social Research Council (ESRC) and the Scottish Graduate School for Social Sciences (SGSSS). The funders had no role in study design, data collection and analysis, decision to publish, or preparation of the manuscript.

The authors declare that they have no relevant or material financial interests that relate to the research described in this paper.”

2. Funding Statement

“These data are from Understanding Society: The UK Household Longitudinal Study, which is led by the Institute for Social and Economic Research at the University of Essex and funded by the Economic and Social Research Council (Grant Number: ES/M008592/1). The data were collected by NatCen Information on how to access the data can be found on the Understanding Society website https://www.understandingsociety.ac.uk/

Data governance was provided by the METADAC data access committee, funded by ESRC, Wellcome, and MRC. (2015-2018: Grant Number MR/N01104X/1 2018-2020: Grant Number ES/S008349/1). Ipsos MORI and Kantar are responsible for conducting the fieldwork for the survey. The UKHLS, supported by the Economic and Social Research Council and multiple Government Departments, is scientifically led by the Institute for Social and Economic Research at the University of Essex.

This PhD research is funded by the Scottish Graduate School of Social Science, as part of the Economic and Social Research Council (ESRC) Doctoral Training Partnership (grant number ES/P000681/1) and Glasgow Caledonian University.”

All page and line numbers refer to the revised manuscript. We will address the comments from each reviewer in turn:

Reviewer 1

1. Abstract:

It is good if the result has statistical result

In response to this comment, and a comment by Reviewer 2, we have re-written the results section of the abstract (page 2, lines 35-46). This includes a statistical result, as stated below: 

“This study reveals a significant decline in mental health, with women exhibiting consistently higher levels of psychological distress than men. Specifically, young people (ages 16-24) experienced a 12.5% greater reduction in mental health compared to individuals over 65 years old. Regression analysis indicated that young adults exhibited a significant increase in psychological distress, with a beta coefficient of 0.85 (p < 0.001), when controlling for other socio-demographic indicators, while loneliness was strongly associated with higher GHQ scores, thus underscoring the critical role of social factors in mental health outcomes”

2. Introduction:

It can be good to also describe the gender gap in mental health in other countries and regions. There could be factors, such as culture, social habit, or race that influence the gender gap in mental health during Covid-19.

Thank you for this comment. We have added a few sentences to the introduction (pg.3, line 65-72), and the gender gap in mental health has been described in a global context and additional references have been included. 

 “Widening gender inequality was a global consequence of the pandemic, with women more likely to experience job loss, increased caring responsibilities, withdrawing from schooling and gender-based violence [8], all of which may contribute to worse mental health outcomes. The underlying factors contributing to the mental health gender gap varies across countries: job loss and caregiving responsibilities are key factors in women's declining mental health in South Korea, Chile, Peru, and Vietnam [9-11], while loneliness and isolation are significant in higher-income countries, particularly among young adults, students, and adolescents [12-15]."

 3. Methods:

 1. It would interest the readers to read more description about the survey.

 2. It is also still important to mention important aspects of the methods such as, sampling techniques used, did all region sampled in the national study.

Thank you for these comments- in response we have revised the methods section to include more information on the study (page 5, lines 98-124). In particular, we have added more information on the sampling methods (page 5, lines 99-106),

“Our data originates from the UK Household Longitudinal Study (UKHLS [19-22]), also known as the 'Understanding Society' study, which is an ongoing panel survey that began in 2009, and collects data from around 40,000 nationally representative households across the UK. The sample comprises: (a) a clustered and stratified general population sample (GPS) proportional to each region of the UK: (b) an ethnic minority boost sample (EMBS) in which participants were sampled from areas of high ethnic minority concentration; (c) an immigrant and minority boost sample (IEMBS), where households were selected from areas of high ethnic minority concentration where at least one member was born outside the UK, or from an ethnic minority group; and (d) a British Household Panel Sample (BHPS) of legacy participants who were part of the previous British Household Panel Survey from 1991-2009 [22].”

, survey content (page 5, lines 106-111) 

“Each wave of data is collected over a 2-year period. For example, the first wave of data collection spans 2009-2011. Participants are interviewed yearly, so wave 2 spans 2010-2012, wave 3 of data collection spans 2011-2013 and so forth. The study collects data on various aspects of people's lives, including their household composition, income, employment, health, education, and social attitudes, and has been used by researchers to investigate gender inequality and mental health during COVID-19 [2,4].”

and the COVID-19 survey (page 5, lines 113-125). 

“The COVID-19 dataset is a sub-set of the UKHLS mainstage surveys that was collected during the COVID-19 pandemic from April 2020 until September 2021. The longitudinal survey began in April 2020, with monthly waves until July 2020, at which point data was collected every 2 months until March 2021. A final wave was fielded in September 2021. Participants of the mainstage yearly Understanding Society surveys were invited to participate in a short web survey during the pandemic (with a telephone option available in May and November 2020 for households where no one was a regular internet user). As participants are sampled from the mainstage Understanding Society samples, the GPS, EMBS, IEMBS and former BHPS members were all eligible for invitation. Eligible participants were 16 years or older in April 2020, and part of an “active” household (i.e. they had completed at least one of the prior two waves of data collection). Participants received £2 for each monthly survey completed, and an additional £10 for the final survey wave. The web survey was fielded by Ipos MORI and the telephone survey was completed by Kantar.”

3. "141,107 observations from 26,335 unique participants" Does the sample size bigger than national study data? It is also different from the total number of sample in table 1 with 141,104 samples, 3 missing data.

We have clarified in our methodology section how this sample compared to the mainstage survey data. 

However, we could not identify the discrepancy you mention in Table 1. 

[page 6, line 127] 

“141,107 observations from 26,335 unique participants”

[page 13, 220]

“1. Unweighted N, total sample= 141,107”

We have investigated the data, and found that sample to be accurate. Discrepancies may be found in Table 1 due to missing data for each demographic characteristic (including, “don’t know”, refusal, proxy and missing responses). 

Reviewer 2

6. Have the authors made all data underlying the findings in their manuscript fully available?

Reviewer #1: Yes

Reviewer #2: No

The secondary data we use in this research is publicly available from the UK Data Service, as stated in the manuscript. 

“ The research data are distributed by the UK Data Service. The mainstage UKHLS survey is available at: https://beta.ukdataservice.ac.uk/datacatalogue/studies/study?id=6614 and the COVID-19 dataset is available at: https://beta.ukdataservice.ac.uk/datacatalogue/studies/study?id=8644” (line 392-396)” 

We have added that this data is publicly available in this section (line 392), but as we did not collect the data ourselves, we do not have the right to distribute it. The data is available publicly on the UK Data Service, and can be accessed by any researchers wishing to replicate this study. 

2. The Abstract:

It would be of interest if the authors include the statistics to give the reader a clearer idea of their findings from the outset.

In response to this and comments from Reviewer 1, we have included a statistical result in the abstract. 

In response to this comment, and a comment by Reviewer 1, we have re-written the results section of the abstract (page 2, lines 35-45). This includes a statistical result, as stated below: 

“This study reveals a significant decline in mental health, with women exhibiting consistently higher levels of psychological distress than men. Specifically, young people (ages 16-24) experienced a 12.5% reduction in mental health compared to individuals over 65 years old. "Regression analysis revealed that young adults experienced a significant rise in psychological distress, indicated by a beta coefficient of 0.85 (p < 0.001), even after accounting for other socio-demographic factors. Additionally, loneliness was found to be strongly linked to higher GHQ scores, highlighting the crucial impact of social factors on mental health. Despite the significant impact of the pandemic on young people's mental health, the analysis highlights a notable recovery to near pre-pandemic levels by September 2021, though age and gender disparities remained pronounced. The mean GHQ score for young adults was 2.56 points higher than that of older adults, reflecting persistent vulnerabilities among younger adults’ mental health in the aftermath of the COVID-19 crisis”. 

3. Results section:

The authors have reported β and p of their random-effects regression models. It would have been informative to see the 95% CIs.

Thank you for this comment. We have taken it into account, and have now included 95% CI’s in Table 2 (p.15-17)

 Another point has to do with the models and the distribution of random effects. While the normality assumption is sometimes hard to ascertain, it is still important to check the extent to which the estimates could be biased if the assumption is not met. In this context, did the authors assess this issue? Did they consider potentially using models with non-normal random effects?

Thank you for your insightful comment regarding the normality assumption of random effects. We agree that assessing the distribution of random effects is crucial to ensure the reliability of the model estimates, however: 

• We have a large sample size (N=141,107), which helps with the robustness of our analysis. As our regression analysis is based on over 100,000 observations, we can be confident that our analysis is rigorous, and that non-normal random effects are not necessary in this case. 

• We have also completed a visual investigation of the data, including a Q-Q plot and histogram of residuals, both of which indicate normality within the distribution of the residuals within each model presented in Table 2. Thus, we can assume normality within the distribution of our data, and be confident that we meet the assumptions for a random-effects model. 

4. Limitation section:

A last point involves the limitations. The only limitation mentioned is that the non-response rate across the survey waves was around 70%. This paper would be stronger if it incorporates other obvious limitations.

Thank you for bringing this to our attention. We have included a more comprehensive limitations section, including some limitations of both the methodology and our analysis (page 23, lines 366-370)

“Random effects models can be less robust to omitted variable bias, which means that the results presented should be interpreted with caution. For this reason, we have focused on change of coefficients from model to model rather than interpreting coefficient size in specific models. We are confident that due to the large sample sizes of the data our estimates remain reliable even with some departures from the normality assumption of the random effects”. 

Once again, we would like to extend our sincere thanks to the reviewers and editors for their time and valuable feedback. We are grateful for the opportunity to enhance the quality of our manuscript through this peer review process, and believe it has substantially improved the quality and clarity of our research. 

Many thanks, 

Mhairi Webster

---

## [Editor Report · Decision Letter 1]

30 Oct 2024

Exploring the Gender Gap in Young Adult Mental Health during COVID-19: Evidence from the UK

PONE-D-24-22134R1

Dear Dr. Webster,

We’re pleased to inform you that your manuscript has been judged scientifically suitable for publication and will be formally accepted for publication once it meets all outstanding technical requirements.

Kind regards,

Kamlesh Kumar Sahu

Academic Editor

PLOS ONE
---

## [Editor Report · Acceptance letter]

26 Nov 2024

PONE-D-24-22134R1 

PLOS ONE

Dear Dr. Webster, 

I'm pleased to inform you that your manuscript has been deemed suitable for publication in PLOS ONE. Congratulations! Your manuscript is now being handed over to our production team.

Kind regards, 

on behalf of

Dr. Kamlesh Kumar Sahu 

Academic Editor

PLOS ONE